

# Findings of the African Combustion Aerosol Collaborative Intercomparison Analysis (ACACIA) Pilot Project to Understand the Optical Properties of Biomass Burning Smoke

Marc N. Fiddler[1], Vaios Moschos[2], Megan M. McRee[3], Abu Sayeed Md Shawon[4], Kyle Gorkowski[4],
James E. Lee[4], Nevil A. Franco[4], Katherine B. Benedict[4], Samir Kattel[3], Chelia Thompson[3], Manvendra
K. Dubey[4], Solomon Bililign[2]

[1]Department of Chemistry, North Carolina A&T State University, Greensboro, NC, 27411, USA
[2]Department of Physics, North Carolina A&T State University, Greensboro, NC, 27411, USA
[3]Department of Applied Sciences and Technology, North Carolina A&T State University, Greensboro, NC, 27411, USA
[4]Earth and Environmental Sciences, Los Alamos National Laboratory, Los Alamos, New Mexico, 87545, USA

*Correspondence to*: Solomon Bililign (bililign@ncat.edu), Manvendra K. Dubey (dubey@lanl.gov)

**Abstract.** Africa is a critical source of biomass burning (BB) aerosols, and its importance is increasing. The African
Combustion Aerosol Collaborative Intercomparison Analysis (ACACIA) Pilot Project set to optically characterize BB aerosol
generated from sub-Saharan African fuels. We used a photoacoustic spectrometer as a reference instrument to determine the
multiple-scattering correction factor $C_\lambda$ for an AE33 aethalometer at three wavelengths, which produced weighted mean values
of $C_{370} = 3.69$, $C_{470} = 5.65$, and $C_{520} = 6.39$. $C_\lambda$ increased with wavelength and $C_{370}$ was statistically independent of the others,
suggesting a single $C_\lambda$ is insufficient, especially in BB scenarios. While a dependence of $C_\lambda$ on burning state was not found, $C_\lambda$
was shown to strongly relate to particle single scattering albedo (SSA, $\omega$). When $C_\lambda$ was plotted against SSA, values slowly
rose at low SSA values, followed by a sharp rise around an SSA of ~0.9; indicating a larger correction needed for less absorbing
aerosol. A number of functions operating on either SSA or $C_\lambda$ were explored and the best function was $-C_\lambda/(1–C_\lambda) = A\omega+B$.
This is an important parametrization of $C_\lambda$ specifically geared towards BB aerosol from African fuels under different aging
states, and is of particular importance for future field work in that continent. An Ångström matrix plot shows that African BB
aerosol can have values more akin to dust, which demonstrates that these fuels are distinct in their wavelength dependence
from more typical BB aerosol. Lastly, we examined the mass extinction and absorbance cross sections for BB aerosol generated
for the same fuels with two different tube furnace setups. Not only is this combustion method flexible, it was found to be
reproducible between labs.




**Graphical Abstract**

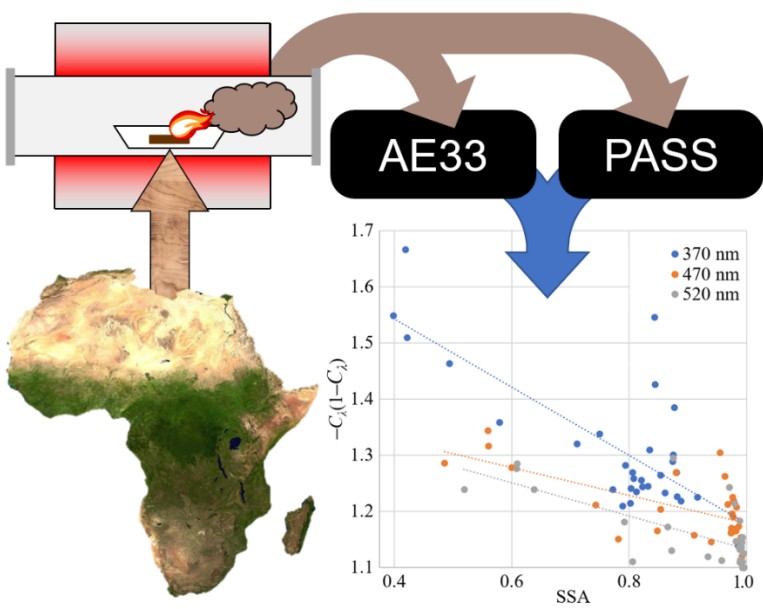

## 1 Introduction

The widespread nature of African biomass burning (BB), as well as the tremendous amounts of primary fine particulate matter (PM$_{2.5}$) and gas-phase emissions that fires produce, have been shown to significantly affect a variety of processes that ultimately impact the Earth's atmospheric composition and chemistry, air quality, water cycle, and climate (Shuman et al., 2022). In Africa, BB is the largest source of aerosols after Saharan dust (Dajuma et al., 2021). Globally, BB-derived aerosols make up most of the primary combustion aerosol emissions, with ~52 % from Africa (Ichoku, 2020; Ichoku et al., 2016). Africa accounts for about 72 % of the total global burned area (van der Werf et al., 2010; Ramo et al., 2021).

Future aerosol emissions in Africa are expected to increase due to rapid economic growth, industrialization, and population growth, as the African population is expected to triple by 2100 (Lamarque et al., 2010; Liousse et al., 2014; Vollset et al., 2020). Recent estimates show that Africa's fire emissions are 31-101 % higher than previous assessments (Ramo et al., 2021). These environmental threats will not only impact Africa's population but will also have a global environmental impact (Ramo et al., 2021). Because the sub-Saharan region of Africa is tropical, BB emissions from this area have a direct and profound influence on the global troposphere. This is due to the presence of a very efficient and deep convective belt that lifts air upwards and leads to transportation of aged, chemically-processed air in the northern and southern subtropics and midlatitudes (Crutzen and Andreae, 1990; Santos et al., 2018; Randel and Jensen, 2013). These factors suggest that Africa may contribute a significant portion of future emissions affecting global atmospheric composition. The influence of Africa's emissions on the Earth's radiative forcing and global air quality are increasingly important, making the need to understand these emissions



urgent. These are formative years for African countries, as the development choices they make now will significantly shape their future emissions and long-term environmental impact.

Optical methods for measuring light absorption in the ultraviolet (UV) and visible (vis) regions of the spectrum are crucial for
understanding the optical properties of BB aerosols (Bond and Bergstrom, 2006; Lack and Cappa, 2010; Moschos et al., 2021). These methods quantify the interaction between light and particles, providing insights into their composition, sources, radiative effects, and atmospheric processes. Intercomparison studies are essential to ensure consistency across various optical instruments and methods, facilitating reliable quantification of aerosol optical properties (Collaud Coen et al., 2004; Müller et al., 2011; Bond et al., 1999; Moosmüller et al., 2009). Such studies might compare in situ methods like photoacoustic
spectrometers (PASS), cavity-attenuated phase shift (CAPS) spectrometers, nephelometer, and filter-based techniques. The latter are widely used for a variety of reasons, however, they are susceptible to scattering artifacts and loading effects (Moschos et al., 2021; Collaud Coen et al., 2010; Ferrero et al., 2024; Kim et al., 2019; Bernardoni et al., 2021; Drinovec et al., 2015; Weingartner et al., 2003). Intercomparisons may help refine correction factors to ensure accurate absorption coefficients, which is especially needed for BB-derived organic aerosols that exhibit complex optical behaviors across wavelengths (Baumgardner
et al., 2012; Bond et al., 1999; Cuesta-Mosquera et al., 2021; Müller et al., 2011).

The dual-spot AE33 model aethalometer is a common filter-based instrument that can be used to measure aerosol absorption properties (Hansen et al., 1984). The AE33 measures light attenuation through aerosol-laden filter spots at multiple wavelengths (370–950 nm), offering insights into wavelength-dependent absorption behaviors. Its figures of merit include mid-cost, ease of operation, broad spectral coverage, automatic loading correction, and ability to take continuous,
unsupervised, real-time measurements. This has led to its extensive use in field and laboratory studies to quantify the light-absorbing characteristics of black carbon (BC) and brown carbon (BrC) aerosols from diverse sources. It is part of the standard instrumentation package used in the DoE ARM Aerosol Observing System (Uin et al., 2019) and us widely used in international monitoring networks, such as the World Meteorological Organization's Global Atmosphere Watch.

However, the method's limitations include potential biases from scattering within the filter material, multiple scattering effects
of the particles, filter loading (i.e. particle shadowing), and the need for wavelength-specific correction factors ($C_\lambda$), which depend on accurate intercomparison with other in situ techniques (Bond et al., 1999; Collaud Coen et al., 2010). The absorption characteristics of combustion particles, especially those containing organic carbon (OC), is influenced by how they get collected on filters – often as liquid coatings rather than solid particles, which can lead to overestimated absorption and biased EC readings during thermal-optical analysis (Subramanian et al., 2007). Weakly-absorbing coatings containing BrC can distort
filter-based measurements like the Particle Soot Absorption Photometer (PSAP) and aethalometer, which have been corrected using side-by-side measurements with photoacoustic devices (Kumar et al., 2022). Measurement uncertainty is also influenced by collection time, spot area, flow rate, and attenuation change (Backman et al., 2017). Using a single wavelength and time independent correction factor ($C$) for aethalometers can yield consistent aerosol heating rate data, but cause biases in the source and speciation apportionments. Applying wavelength-dependent $C_\lambda$ results in more precise and consistent outcomes, while a
time-dependent correction ($C_t$), using a separate measurement such as a multi-wavelength absorption analyzer (MWAA), has



minimal effect on heating rates (Ferrero et al., 2021; Yus-Díez et al., 2021). Filter loading can cause errors of up to ±50 %, depending on aerosol properties (Drinovec et al., 2017). A number of other factors can significantly affect aethalometer measurements, including filter equilibration, particle penetration depth, and aerosol type, especially for complex aerosols (Arnott et al., 2005; Drinovec et al., 2017). Despite these limitations, the aethalometer remains a valuable tool for aerosol

optical characterization, particularly when combined with complementary methods to account for biases and improve data reliability (Drinovec et al., 2015; Moschos et al., 2024; Moschos et al., 2021; Cuesta-Mosquera et al., 2021; Saturno et al., 2017; Yus-Díez et al., 2021; Ferrero et al., 2024; Weingartner et al., 2003; Arnott et al., 2005). However, there have only been a few instances where the AE33 has been applied exclusively to ambient BB aerosol (Schmid et al., 2006).

Such aethalometer absorption measurements have been applied to BB aerosols generated in a laboratory environment, which

enables the control of fuel type, exposure to volatile organic compounds (VOCs) and oxidants, and combustion state (McRee et al., 2024; Moschos et al., 2024). Previous work in our laboratory has used a tube furnace for generating BB aerosol, which has allowed a high degree of flexibility in accessing different combustion states of a fuel (Pokhrel et al., 2021a). Owing to its flexibility and commercial availability, the use of this combustion setup has been growing (Benedict et al., 2024; Hoffer et al., 2017).

The African Combustion Aerosol Collaborative Intercomparison Analysis (ACACIA) Pilot Project's aim was to optically characterize BB aerosol generated from African fuels, correct the AE33 aethalometer at multiple wavelengths, and parametrize the correction factors in well-controlled and characterized conditions. The AE33 is likely to see use in long-term monitoring in future field campaigns in Africa, which makes determining $C_\lambda$ for these BB aerosol essential. This was accomplished by conducting laboratory studies at Los Alamos National Laboratory (LANL) and North Carolina A&T State University (NCAT),

where aethalometer measurements were compared to in situ optical measurements. Mass cross sections of BB aerosol from these under-studied fuels will be determined along with their wavelength dependence. Additionally, we will examine the wavelength dependence of scattering and absorption and place it in the context of other BB aerosol measurements. Lastly, results from combusting the same fuels with two different tube furnace systems will be compared to assess the general applicability of this flexible means of producing BB aerosol.

**2 Methods**

**2.1 Fuels**

African biomass fuels obtained directly from Ethiopia, Botswana, and a few additional fuels that were collected near LANL. These fuels were selected to be representative of those burned for domestic use in Africa and wildland fires in New Mexico (NM). A thorough description of each fuel is given in Text S1. Fuel samples were stored in a fume hood to dry and their

moisture content was measured (PCE-MA 50x, PCE Instruments). All fuels were found to have a moisture content of <10 %, and were considered dry (McRee et al., 2024). The spatial extent and use of each fuel are described in SI and they are listed in Table 1. Botswana samples were collected in Palapye.





**Table 1.** Fuels used for these experiments.

| Common Name | Scientific Name | Source |
|---|---|---|
| Bahir Zaf/Eucalyptus | *Eucalyptus camaldulensis* | Ethiopia |
| Acacia | *Acacia abyssinica* | Botswana |
| Wanza | *Cordia africana* | Ethiopia |
| Mopane/balsam tree | *Colophospermum mopane* | Botswana |
| Ethiopian cow dung | *n/a* | Ethiopia |
| African fountain grass/ Savannah grass | *unknown* | Botswana |
| Ponderosa Pine | *Pinus ponderosa* | Los Alamos, NM |
| Blue Grama | *Bouteloua gracilis* | Los Alamos, NM |

## 2.2 LANL Combustion and Measurement System

A total of 42 burns were conducted during April of 2023 at LANL in the Center for Aerosol and trace-gas Forensic Experiments (CAFE; Los Alamos, NM, elevation ~1900m). The system is described extensively elsewhere (Benedict et al., 2024), and the specific configuration used in this work is shown in Fig. 1. Briefly, preweighed samples of fuel (0.47-2.16 g) were placed in a quartz boat and inserted into the capped quartz tube of the tube furnace (TS1 12/60/150-120SN, Carbolite Gero), where combustion took place. Several furnace temperatures were tested (see Table S1) to simulate a variety of burning conditions. A zero-air generator (T701, Teledyne) produced a flow of dry particle free air through the system. Flow from the furnace passed by an oxygen sensor (Oxy-Flex-1-H) and was sent to a custom 34 L stainless steel mixing tank to allow for cooling, dilution, and extended sampling (15–20 min). The fire-integrated modified combustion efficiency (MCE) was derived from cavity ring-down measurements of CO and $CO_2$ measurements (G2401, Picarro). Several instruments were used to measure speciated NO and $NO_2$ by UV absorption (405, 2B Tech) and chemiluminescence (T200, Teledyne).



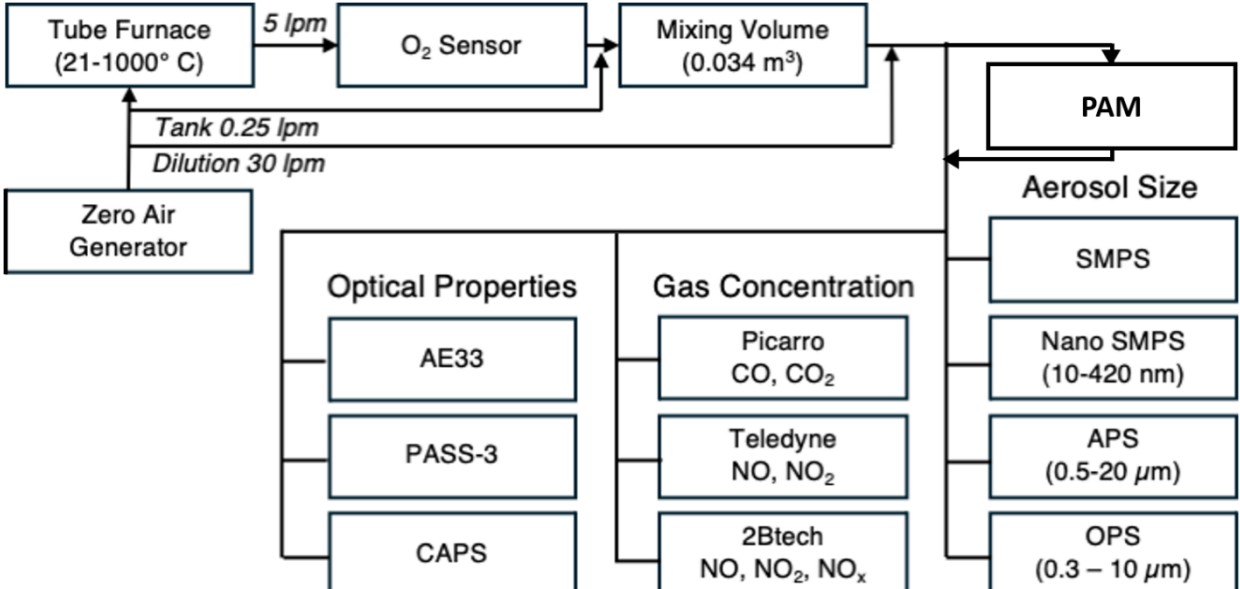

**Figure 1. A diagram of the system at LANL used for generating and characterizing BB aerosol.**

Multiple methods were used to measure the wavelength-dependent light absorption properties of aerosol produced from combusting biomass fuels. A three-wavelength photoacoustic soot spectrometer (PASS-3, Droplet Measurement

Technologies) was used to measure absorption and scattering coefficients at 405 and 532 nm (Arnott et al., 1999); the 781 nm channel was degraded and not used here. This instrument has been applied to absorbing aerosol (McMeeking et al., 2014; Lewis et al., 2008; Flowers et al., 2010) and its performance has been previously characterized (Nakayama et al., 2015). The scattering and absorption coefficients have an estimated uncertainty of 40 % (Benedict et al., 2024; Virkkula et al., 2005). The aethalometer is described in Sect. 2.4. The aerosol size distribution was measured using a combination of scanning mobility

particle sizer (SMPS, models 3082 and 3750), nano-SMPS (model 3910), aerodynamic particle sizer (APS, model 3320), and optical particle sizer (OPS, model 3330); all from TSI, Inc. Size distributions were corrected for multiple charging and diffusion using in-software calculations (Drinovec et al., 2015). This resulted in a combined size distribution between 10 nm and 10 μm, which was used to determine the particle at mass 2-minute intervals (see Sect. 2.4).

## 2.3 NCAT Combustion and Measurement System

The combustion system, smog chamber, and particle and gas measurement systems at NCAT have been previously characterized (Smith et al., 2019), which includes lamp spectral characteristics and particle and gas lifetimes. The current configuration is shown in Fig. 1 of McRee et al. (2024), where it is also described extensively. Briefly, fuels were burned in a tube furnace (Carbolite Gero, HST120300-120SN) and the smoke was introduced through a 2.5-μm cut point cyclone into a 9 m³ Teflon smog chamber at NCAT. Particle distributions were measured by an SMPS (3080 and 3787, TSI, Inc.). $O_3$ (211, 2B

Technologies), $NO_x$, CO, and $CO_2$ (42iQ, 48iQ, and 410iQ, respectively, from Thermo Scientific) were also measured for all



burns. The last two were used to derive MCE from the change in CO and CO$_2$ from before the experiment and the peak value. By setting the tube furnace to temperature of 450 °C, the resulting MCE values from all experiments were < 0.9, which is indicative of smoldering-dominated conditions that produce spherical particles that are predominantly OA (Pokhrel et al., 2021b). Only fresh emissions without the addition of more oxidant were studied in this work. The chamber was kept dry for

these experiments, where the RH was 0–10 %.

Light extinction was measured at 550 nm using a home-built cavity ring-down spectrometer (CRDS), which has been described previously (McRee et al., 2024; Smith et al., 2019). This instrument was operated with a 10:1 dilution ratio using a mixing jar. While a Nephelometer was also used, aethalometer-based absorption values were found to be more reliable for determining cross sections (including scattering, by subtraction) and SSA.

**2.4 Aethalometry**

A dual-spot aethalometer (AE33, Magee Scientific) was used to quantify aerosol light absorption. The AE33 system measures the light transmission through aerosol-loaded spots on a filter tape ($I$), compared to the transmission through a filter tape spot that has clean air passed through it as a reference ($I_0$). It simultaneously measures the light attenuation at seven wavelengths ($\lambda$) from the near-infrared (950 nm) to near-UV (370 nm). The optical attenuation ($ATN_\lambda$) was found at 370, 470, and 520 nm,

which was defined as $ATN_\lambda = -100 \cdot \ln(I_\lambda/I_{0,\lambda})$. For NCAT burns, the AE33 measurements and data analysis were described in detail in Moschos et al. (2024). For LANL burns, we determined the attenuation coefficient, $\alpha_{atn,\lambda}$ (Mm$^{-1}$) for optically-absorbing aerosols by evaluating the rate of change of light attenuation passing through the particle-laden filter, using the formula:

$$\alpha_{atn,\lambda} = D \cdot S \cdot (\Delta ATN_\lambda/100)/(F_{in} \cdot \Delta t), \tag{1}$$

where $S$ represents the spot area, $F_{in}$ is the aerosol flow rate (6 L min$^{-1}$), and $\Delta t$ is the time (1 min resolution). Particle-free chamber air, made by placing a HEPA filter at the inlet, had no contribution to $\alpha_{atn,\lambda}$ (Moschos et al., 2024). High aerosol mass concentrations required dilution with zero air using a dilution jar during experiments in both laboratories, where the dilution ratio ($D$) was 5, and a lower flow rate of ~1.44 L min$^{-1}$ was used to produce a relatively small $\Delta ATN/\Delta t$. Only $ATN$ values <40 were included, where the dual-spot automatic loading-correction was reliable (based on our test measurements and previous

work (Drinovec et al., 2015)), which is why a loading term is not included in Eq. 1 (Ferrero et al., 2024). For each burn, we allowed time for the signal to stabilize (i.e. produce relatively constant $\Delta ATN/\Delta t$) and recorded data at 1 min averages. A slight cross-sensitivity between scattering and absorption has been observed in the AE33. Ammonium sulfate particles exhibit an apparent absorption of 1–2.3 % of scattering, depending on ATN, wavelength, and average particle size (Drinovec et al., 2015). A correction for this effect was not incorporated into this work, since it would not be a significant source of systematic error.

The AE33 correction factor $C_\lambda$ was calculated by comparing the $\alpha_{atn,\lambda}$ from the aethalometer and absorption coefficient $\alpha_{abs,\lambda}$ (Mm$^{-1}$) measured or derived from a multi-wavelength PASS (Moschos et al., 2024; Moschos et al., 2021; Collaud Coen et al., 2010; Müller et al., 2011; Weingartner et al., 2003). $C_\lambda$ is calculated using Eq. 2. Since the PASS operates at other wavelengths (405 and 532 nm), $\alpha_{abs,\lambda}$ at AE33 wavelengths of 370, 470, and 520 nm were extrapolated by assuming a power-law relationship



(i.e. deriving an Ångström absorption exponent (AAE)). This extrapolation is shown in Fig. S1 for burn 17. The extinction
coefficients $\alpha_{\text{ext},\lambda}$ (Mm$^{-1}$), and scattering coefficients $\alpha_{\text{scat},\lambda}$ (Mm$^{-1}$) are similarly extrapolated from their Ångström extinction
exponent (AEE) and Ångström scattering exponent (ASE), respectively. The Model 8060 filter tape was used in this work,
which is important to note, since $C_\lambda$ is influenced by filter type.

$$C_\lambda = \alpha_{\text{atn},\lambda}/\alpha_{\text{abs},\lambda} \tag{2}$$

Cross sections were calculated using the particle mass loading $M$ (µg m$^{-3}$). This was derived from the size distribution, which
was measured with either an SMPS (NCAT) or combination of OPS/nano-SMPS (LANL), and used a previously measured
particle density $\rho$ of 1.2 g cm$^{-3}$ (Pokhrel et al., 2021b). Since this was only applied to burns that were dominated by smoldering
combustion (i.e. MCE ≤ 0.91), particles could be regarded as spherical (Pokhrel et al., 2021b). The mass absorption cross-
section (MAC, m$^2$ g$^{-1}$) is calculated using Eq. 3, with mass scattering and extinction cross-sections (MSC and MEC,
respectively) being calculated similarly. LANL experiments had relatively short particle lifetimes in their mixing volume, and
size distributions were measured every two minutes. To match the one-minute averages of optical measurements and account
for changes over time, the mass loading was extrapolated between measured values. The AE33 does not exhibit a significant
dependence of $C_\lambda$ on particle size (Drinovec et al., 2015).

$$\text{MAC}_\lambda = \alpha_{\text{abs},\lambda}/M \tag{3}$$

The SSA is calculated according to Eq. 4, depending on the setup. At LANL, SSA values were based on extrapolated scattering
and absorption from the PASS, while NCAT values used direct extinction (CRDS) and absorption (aethalometer)
measurements.

$$SSA = \frac{MSC}{MSC+MAC} = \frac{MEC-MAC}{MEC} \tag{4}$$

Aethalometer measurements have a time resolution of 1 minute. Since PASS measurements were recorded every two seconds,
these measurements were averaged for one minute before being used to compare the aethalometer. Depending on the
experiment, this resulted in 4 to 22 aethalometer measurements (an average of 9 measurements) per burn. Between 1 and 6
burns were performed for each African fuel and 2 for each North American fuel. Extrapolated values for $\alpha_{\text{abs}}$ and $C_\lambda$ were
determined for each minute and presented values were averaged over the course of each burn. Errors are the standard deviation
of 1-minute values within each experiment. A few burns had absorption measurements that varied too much over the course
of the experiment. Burns that had an average relative standard deviation (RSD) that exceed 10 % for $C_\lambda$ and 5 % for SSA were
rejected for further analysis. This resulted in the rejection of burn #21 (savanna grass), #23 (wanza), and #34 (eucalyptus). The
remaining 31 burns had an average RSD of 6.1 % for $C_\lambda$ and 0.4 % for SSA over all wavelengths.

## 3 $C_\lambda$ determination and parametrization

Characterizing the wavelength-dependent properties of PM is important for understanding their role in radiative forcing and
subsequent climate effects at the regional and global scale (Bond et al., 2013; Kirchstetter et al., 2004). When using a filter-
based instrument, there are three main factors that should be addressed (Bond et al., 1999; Liousse et al., 1993; Petzold et al.,





1997): 1) filter fibers increase attenuation by multiple scattering (which increases optical path), 2) aerosols embedded in the filter increase light attenuation because of scattering, and 3) as light absorbing particles collect in the filter, the attenuation gradually increases (reduces optical path for loaded filter). The correction factor $C_\lambda$ accounts for multiple scattering of light within the filter and the resulting enhancement of the optical path of aerosol particles that load up the filter (Ferrero et al.,

2021). Previous studies suggest that a single correction factor at each aethalometer wavelength may not be sufficient, because it has been shown to vary based on type of aethalometer, material of the filter, seasonal or diurnal patterns that are produced by PM compositional changes, and wavelength (Bernardoni et al., 2021; Yus-Díez et al., 2021). To investigate dependencies of the correction factor in this study we analyzed its relationship with burning state and SSA.

### 3.1 Average $C_\lambda$ and wavelength dependence

Measurements for each burn are presented in Table S1 with burn experiment number, fuel, temperature of the tube furnace, MCE, AAE, ASE and AEE from 405 to 532 nm using PASS measurements, average $C_\lambda$ at three wavelengths, and extrapolated SSA values at these wavelengths. Median and weighted mean values of $C_\lambda$ of all fuels in this work are presented in Table 2, along with previous measurements of $C_\lambda$ that used an AE33 with model 8060 filter tapes. An outlier of $C_{470}$ of 0.93 was excluded from this analysis to be consistent with later sections (3.3). Regardless if it is represented as a median or weighted

mean, values of $C_\lambda$ increase with increasing wavelength in this work. A two-tailed, unpaired statistical analysis of $C_\lambda$ with unequal variance showed that $C_{370}$-$C_{470}$ had a P-value of $4.5 \times 10^{-8}$, $C_{470}$-$C_{520}$ had a P-value of 0.10, and $C_{370}$-$C_{520}$ had a P-value of $1.0 \times 10^{-7}$. While $C_{470}$ could not be distinguished from $C_{520}$ at 95 % confidence, $C_{370}$ was distinct from the other two. Even at a relatively small wavelength range, a wavelength dependence of $C_\lambda$ is observed in this work.

**Table 2.** Mean $C_\lambda$ for this work and previous works that used an AE33 with model 8060 filter tapes. Errors are 1 standard deviation that represent run-to-run variability, which was greater than the propagated weighted mean standard deviation, or are defined in their respective works. Yus-Díez et al. (2021) report median values.

| Reference | Sample Type | $C_{370}$ | $C_{470}$ | $C_{520}$ |
|---|---|---|---|---|
| Median, this work | Chamber BB OA | 4.71 | 6.17 | 8.35 |
| Weighted Mean, this work | Chamber BB OA | 3.69±1.14 | 5.65±1.28 | 6.39±2.09 |
| Moschos et al., 2021 | Ambient EC/OC | 2.48±0.39 | 2.40±0.36 | 2.34±0.36 |
| Moschos et al., 2024 | Chamber BB OA | 4.1 | 3.20 | 2.9 |
| Valentini et al., 2020 | Urban Background | | 2.66 | |
| Yus-Díez et al., 2021 UniMI Polar Photometer | Urban Background | 3.36 | 3.26 | 3.22 |
| | Regional Background | 2.68 | 2.67 | 2.72 |
| | Mountaintop | 3.47 | 3.48 | 3.58 |
| Yus-Díez et al., 2021 MAAP | Urban Background | 2.82 | 2.78 | 2.75 |
| | Regional Background | 2.32 | 2.33 | 2.42 |
| | Mountaintop | 2.82 | 2.85 | 2.91 |





The wavelength dependence of $C_\lambda$ has not generally been previously observed or pursued. The values found in this work are
somewhat larger than those determined in an urban setting (Valentini et al., 2020). In Moschos et al. (2021), an AE33/MWAA
comparison was carried out on ambient quartz filter samples from a year-long sampling campaign in Switzerland, where the
aerosols consisted of both elemental and organic carbon. All aethalometer wavelengths were examined in that work, which
resulted in a largely wavelength-independent $C_\lambda$ of ~2.3, as shown in Table 2. Previous work with the AE31 found either no
wavelength dependence (a constant $C_\lambda$ value of 2.14) (Segura et al., 2014), a slight decrease with wavelength that was not
statistically significant (Bernardoni et al., 2021), or a strong indication of wavelength independence (Weingartner et al., 2003).
$C_\lambda$ can also strongly depend on the filter material used, with values for the M8020 filter tapes used in current AE33 instruments
ranging from 2.57 to 4.24 (Drinovec et al., 2015; Yus-Díez et al., 2021). For NCAT-chamber African BB-OAs (smoldering
burns; no elemental carbon present), Moschos et al. (2024) found the $C_\lambda$ values to be wavelength-dependent: $C_{370} = 4.1$, $C_{470}$
$= 3.2$, $C_{520} = 2.9$, and $C_{590} = 2.0$; exhibiting a decreasing correction factor with increasing wavelength, but no systematic
variation between fuels or aging conditions. Focusing on mountaintop measurements that were influenced by BB, Yus-Díez
et al. (2021) showed a statistically significant increase between $C_{470}$ and $C_{520}$, but no signifiant difference between $C_{370}$ and
$C_{470}$. While our values were slightly higher, our trends are in agreement.

### 3.2 $C_\lambda$ Parametrization for burning state

Long- and short-term field campaigns frequently measure CO and $CO_2$, owing to the stability, accuracy, and utility of these
monitors. Multiple simultaneous optical measurements are more demanding and less common, so a method to determine $C_\lambda$
that is not optically-based would be attractive. Since CO and $CO_2$ can be used to calculate the fire-integrated MCE, and MCE
is linked to the burning state, we examined the dependence of $C_\lambda$ on the burning state. The MCE has been used to establish
whether the combustion is in a smoldering-dominated, flaming-dominated, or mixed combustion regime (Pokhrel et al., 2021b;
Yokelson et al., 1997). A plot of $C_\lambda$ against MCE at 370, 470, and 520 nm is presented in Fig. 2. An unweighted least squares
linear fit was also done for each wavelength. The resulting $R^2$ values vary between 0.01 and 0.12, which is to say that less than
12 % of variations in $C_\lambda$ can be attributed to differences in MCE. An analysis of the fitted slope and its error produced was
compared to zero, which resulted in p-values of 0.62, 0.48, and 0.12 for 370, 470, and 520 nm measurements, respectively.
That is, none of the slopes were statistically different from zero at 95 % confidence. While a correction factor that does not
rely on some other optical measurements would be desirable, it does not appear that $C_\lambda$ exhibits an MCE dependence.



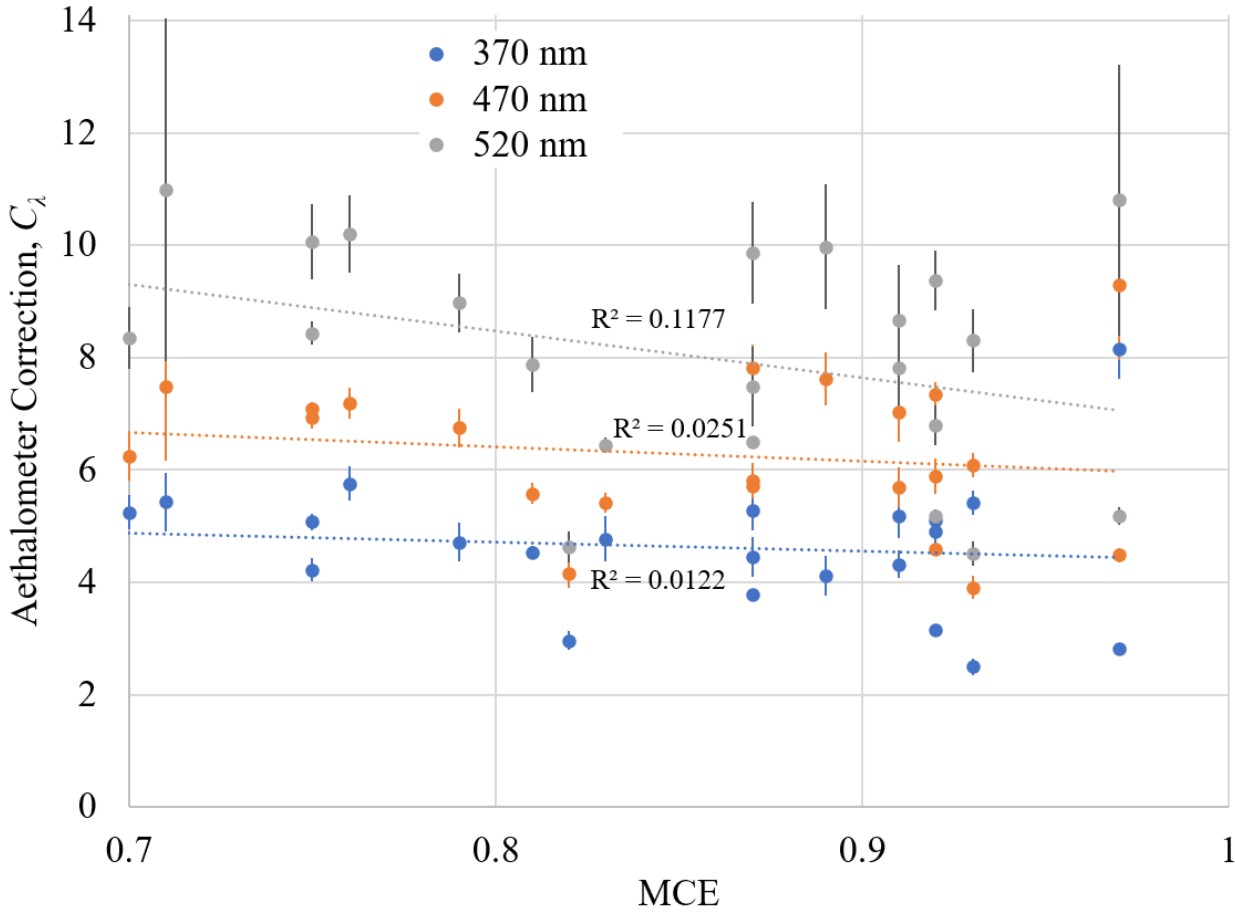

**Figure 2. The aethalometer correction factor $C_\lambda$ at three wavelengths plotted against MCE. Experiments where there was not sufficient data to calculate MCE, as well as PAM oxidation experiments, were not included.**

### 3.3 $C_\lambda$ Parametrization Based on SSA

Compositional differences in the particle phase have led to complex and non-linear relationships between $\alpha_{atn,\lambda}$ and $\alpha_{abs,\lambda}$ for aethalometer measurements. These differences have been previously attributed to filter loading effects, where the loading varies with seasonality due to compositional changes (Virkkula et al., 2007). Fresh aerosols, typically rich in BC from biomass burning or traffic emissions, tend to produce stronger loading effects compared to aged aerosols that have undergone chemical processing (Weingartner et al., 2003). Beyond loading effects, several studies have explored a more intrinsic link between $C_\lambda$ and SSA. Notably, Schmid et al. (2006) and Collaud Coen et al. (2010) first proposed this relationship, which was later expanded upon by Ferrero et al. (2024), who demonstrated that SSA could serve as an indicator of aerosol loading effects and internal mixing state.




In our study, we observed that $C_\lambda$ consistently exhibited lower values at lower SSA across all three aethalometer wavelengths, as demonstrated in Fig. 3, in agreement with previous findings. This relationship was also evident in experiments using artificially aged BB aerosol oxidized in a PAM reactor, indicating the relevance of the parameterization for aged aerosol.

**Figure 3. The aethalometer correction factor $C_\lambda$ plotted against SSA at three wavelengths. PAM oxidation experiments were included. Results of a linear fit are shown.**

Several functions were fit to the resulting plots using unweighted least squares fitting. These functions are shown in Table 3 along with the resulting $R^2$ for each fit. An outlier of $C_{470}$ of 0.93 was excluded from this analysis, since some fit functions are incompatible with values below 1 (see Fig. S1). While measurement variability limited the performance of fit functions, meaningful relationships with SSA were observed. $C_{370}$ showed the strongest dependence, where SSA determined 43 % to 57 % of $C_{370}$ variability, depending on the fit. $C_{470}$ showed the weakest dependence, with only 28 % to 43 % attributed to SSA, while $C_{520}$ exhibited about 38 % to 40 % variability dependence. We also tested the functional form proposed by Schmid et al.





(2006) and later used by Yus-Díez et al. (2021), which is $C_\lambda = A(\omega/(1-\omega))+B$, where $\omega$ is the SSA. Urban, rural, and

mountaintop observations in Northeastern Spain showed a relatively stable $C_{637}$ of ~2.4 followed by a sharp rise at high SSA values, where in $C_{637}$ was ~7.5 at an SSA of ~1 (Yus-Díez et al., 2021). Their median $C_{637}$ was 2.23 to 2.51 depending on the site. At lower SSA values, our $C_{370}$ had similar values and peaked similarly at high SSA values. At 470 and 520 nm, our $C_\lambda$ values were 4–5 at lower SSA values, and peaked higher at high SSA values, with $C_\lambda$ being 9–11. A fit of the above function to $C_{637}$ led to values of A = 1.96 and B = 3.0 for their regional background site and A = 1.82 and B = 4.9 for their mountaintop

observations. They did not perform such a fit at shorter wavelengths. Our data was fit to this function and produced a relatively good fit for 470 nm values, but owing to the degree of scatter in our observations, this function did not behave particularly well at other wavelengths (see Table 3).

**Table 3.** Fit functions applied to the plot of $C_\lambda$ against SSA, shown in Fig. 3, the $R^2$ and the Chi squared ($X^2$) for each fit.

| Function | Form | $R^2$ 370 | 470 | 520 | $X^2$ 370 | 470 | 520 |
|---|---|---|---|---|---|---|---|
| Linear | $C_\lambda = A\omega+B$ | 0.467 | 0.311 | 0.393 | 4.12 | 5.21 | 9.73 |
| Polynomial | $C_\lambda = A\omega^2+B\omega+D$ | 0.491 | 0.322 | 0.396 | 3.99 | 5.10 | 9.71 |
| Log | $C_\lambda = A\cdot\ln(\omega)+B$ | 0.446 | 0.318 | 0.386 | 4.33 | 5.14 | 9.88 |
| Exponential | $C_\lambda = Ae^{B\omega}$ | 0.485 | 0.305 | 0.396 | 4.11 | 5.39 | 10.04 |
| Power Law | $C_\lambda = A\omega^B$ | 0.463 | 0.314 | 0.394 | 4.30 | 5.29 | 10.06 |
| Schmid/Yus-Díez | $C_\lambda = A\omega/(1-\omega)+B$ | 0.433 | 0.329 | 0.379 | 5.06 | 5.53 | 11.08 |
| | $C_\lambda = -A/\ln(\omega)+B$ | 0.434 | 0.330 | 0.379 | 5.06 | 5.53 | 11.07 |
| Arctangent | $C_\lambda = A\cdot\arctan(\omega)+B$ | 0.453 | 0.317 | 0.389 | 4.26 | 5.15 | 9.81 |
| | $C_\lambda = A\cdot e^{(\omega-1)}/(1-\omega)+B$ | 0.430 | 0.329 | 0.379 | 5.09 | 5.54 | 11.08 |
| Nested Exp. | $C_\lambda = A\cdot e^{e^\omega}+B$ | 0.511 | 0.284 | 0.395 | 3.85 | 5.51 | 9.85 |
| | $-C_\lambda/(1-C_\lambda) = A\omega+B$ | 0.576 | 0.429 | 0.476 | 0.14 | 0.04 | 0.05 |
| | $1/\ln(C_\lambda) = A\omega+B$ | 0.574 | 0.417 | 0.470 | 0.37 | 0.16 | 0.20 |
| | $\arctan(C_\lambda) = A\omega+B$ | 0.562 | 0.408 | 0.469 | 0.03 | 0.02 | 0.02 |

A number of other functions that exhibit sharp rises at higher values were also examined in Table 3. Of those functions examined that fit a function of SSA, the second order polynomial performed the best, though this is to be expected with more fit parameters. The nested exponential function performed had the second highest average $R^2$, though it had the worst $R^2$ for $C_{470}$ of those tested. This was followed by the exponential function, and then linear and power law fits performing comparably. Other functions, including several sigmoid functions, did not perform as well, on average. A linear fit to several functions that

manipulate $C_\lambda$ were also examined, which performed better than those that only adjust SSA. Along with $R^2$, Table 3 shows how well each function fit the data by calculating the goodness of fit, i.e. Chi squared ($X^2$). Of those, $-C_\lambda/(1-C_\lambda) = A\omega+B$ performed the best, where 49 % of the variability of the Y-term depended on SSA and this equation had had the second lowest $X^2$. Solving for $C_\lambda$, this would have the form $C_\lambda = (A\omega+B)/(A\omega+B-1)$, which has a strong potential for future use in SSA-



based correction schemes, particularly where filter loading effects are minimized and optical properties dominate the
measurement bias. While arctan($C_\lambda$) had slightly lower $X^2$ values, the $R^2$ was  also lower, so it exhibited a weaker $C_\lambda$
dependence. The fit parameters of the best-performing fit equations that manipulate SSA (the nested exponential) and $C_\lambda$ ($-$
$C_\lambda/(1-C_\lambda)$) are shown in Table 4 and are plotted in Fig. S2.

**Table 4.** The resulting fit parameters of functions applied to $C_\lambda$ and SSA for the best overall fits. Parameters A and B are in
the function and values are given at each wavelength in this study.

|  | $C_{370}$ | | $C_{470}$ | | $C_{520}$ | |
| --- | --- | --- | --- | --- | --- | --- |
| Function | A | B | A | B | A | B |
| $C_\lambda = A e^{e^\omega} + B$ | 0.3502 | 1.3030 | 0.2180 | 3.5103 | 0.4229 | 2.6202 |
| $-C_\lambda/(1-C_\lambda) = A\omega+B$ | -0.6074 | 1.7855 | -0.2489 | 1.4279 | -0.2972 | 1.4296 |

## 4 Wavelength Dependence of sub-Saharan BB Aerosol

Characterizing the wavelength dependent properties of PM is important for understanding their role in radiative forcing and
subsequent climate effects at the regional and global scale (Bond et al., 2013; Kirchstetter et al., 2004). AAE describes the
wavelength dependence of aerosol absorption, which assumes a power-law relationship (Moosmüller et al., 2011). Plots of
ASE vs AAE have been used to differentiate different particle types and the source of those particles. Measurements in this
work are presented on this so-called Ångström matrix plot in Fig. 4 with source labels are derived from Cazorla et al. (2013).
Also included are previous observations of African fuels for fresh and photo-aged emissions (McRee et al., 2024).





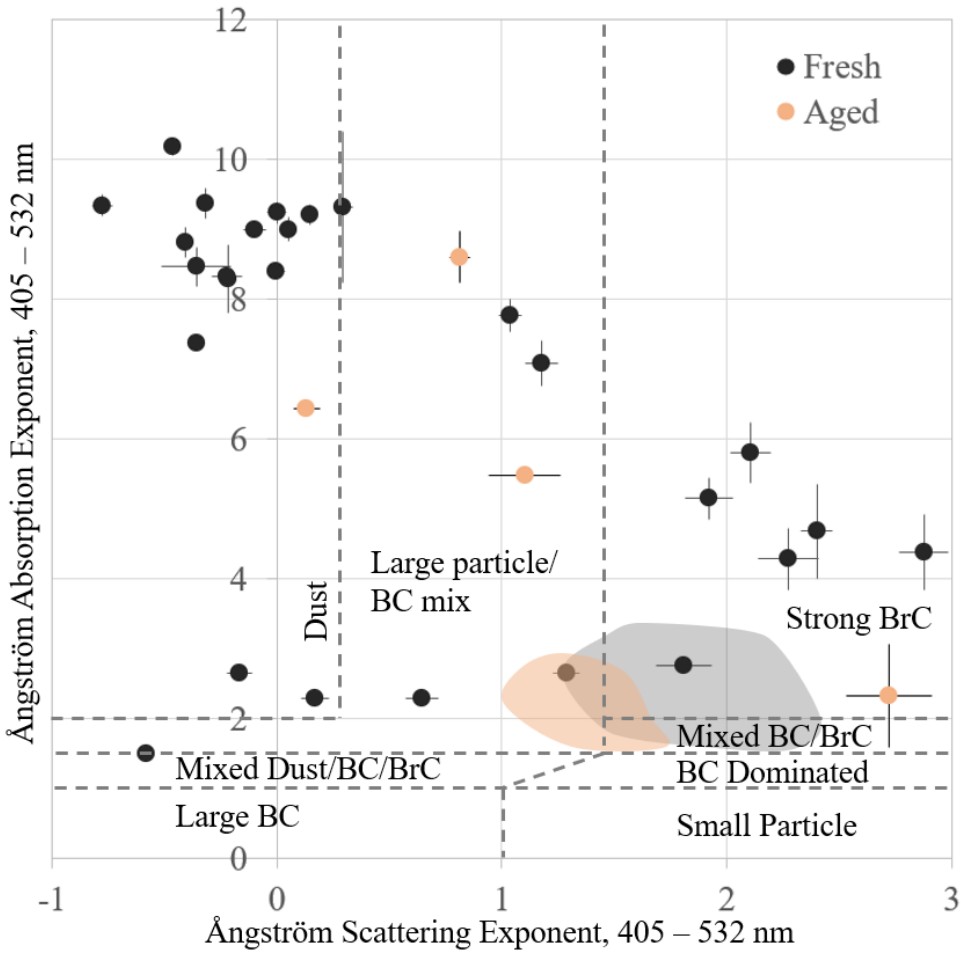

**Figure 4. The Ångström matrix plot (ASE vs AAE) for fresh and PAM-aged observations in this work and previous work on similar fuels (in shaded areas) (McRee et al., 2024).**

Values observed in this work were wide ranging. ASE ranged from -0.78 to 2.88 with an average of 0.61, where negative values showed enhanced scattering at 523 nm, as opposed to 405 nm. AAE ranged from 1.50 to 10.18, with an average of 6.47. AAE values of ~1.0 have been observed for submicron refractory BC-rich aerosol produced from fossil fuel combustion, while larger AAE values up to 9.5 have been observed and attributed to larger BrC particles from BB or absorbing dust (Liu et al., 2018; Kirchstetter et al., 2004; Lack and Langridge, 2013). Cazorla et al. (2013) showed that organic-dominated PM had AAE

and ASE values both ranging from 1.5 to 3. BB observations during a wildfire season by Ponczek et al. (2022) found maximum values of 3.58 for AAE and 2.31 for ASE. Likewise, laboratory studies by Zhang et al. (2020) found AAE values from 1.17 to 2.92, while BrC-laden BB particles by Martinsson et al. (2015) found AAE values of 2.5–2.7. Ambient measurements in NE Spain found average AAE values between 1.12 and 1.35 depending on the site (Yus-Díez et al., 2021). While these literature values were close to the range observed for similar African fuels and previous studies (McRee et al., 2024), as indicated in

Fig. 4, the range of values in this work is clearly much greater and even slightly exceeding previously observed values of AAE,





being more akin to dust observations. This clearly shows that BB aerosol from African fuel sources are distinct in their optical properties.

Previous measurements of BB particles from similar fuels showed that the range of values for both AAE and ASE decreased upon photoaging, as well as with dark aging and dark aging with additional nitrate radical (McRee et al., 2024). This

demonstrated that both processes reduced the wavelength dependence of scattering and absorption.

## 5 Comparison of Combustion Systems

The flexibility of the tube furnace over other combustion systems has been demonstrated (Pokhrel et al., 2021b). If we hope for more wide-spread use, however, its reproducibility has to also be examined. To this end, we have compared the results of smoldering-dominated combustion of similar fuels using both systems. Cross sections were calculated using Eq. 2,

had units of $m^2\ g^{-1}$, and were measured under dry conditions; <10 % at both NCAT and LANL. Both SSA values are derived from (MEC–MAC)/MEC. NCAT measurements are from McRee et al. (2024) and use a MWAA instrument for correcting the aethalometer measurements. Results from fresh emissions are presented in Table 5. Like in McRee et al. (2024), averages and standard deviations in this table were between repeated experiments (i.e., the run-to-run variability) and did not propagate the standard deviation of individual experiments. Values for individual experiments are in Table S1. Burn 18 for Wanza was not

included because of its unusually large MEC.

**Table 5.** Comparison of optical properties for fresh and photo-aged emissions from various fuels, carried out at LANL and NCAT (McRee et al., 2024). All cross sections are in $m^2\ g^{-1}$. MCEs with a * had one or more values were not recorded because of a lack of CO and $CO_2$ measurements during those experiments, but should be smoldering-dominated, since the furnace temperature was 450 °C at both LANL and NCAT.

| State | Fuel | MCE | NCAT; 550 nm MEC (CRDS) | MAC (AE33) | SSA | n | MCE | LANL; 532 nm MEC (PASS) | MAC (PASS) | SSA | n | P-Value MEC | MAC | SSA |
|---|---|---|---|---|---|---|---|---|---|---|---|---|---|---|
| Fresh | Acacia | * | 5.01 ±0.80 | 0.220 ±0.078 | 0.956 ±0.017 | 2 | 0.84 ±0.08* | 6.63 ±2.73 | 0.582 ±0.961 | 0.912 ±0.149 | 4 | 0.333 | 0.509 | 0.604 |
| | Wanza | 0.84 | 8.43 ±1.34 | 0.394 ±0.133 | 0.953 ±0.017 | 2 | * | 4.45 ±0.09 | 0.042 ±0.004 | 0.991 ±0.001 | 2 | 0.149 | 0.167 | 0.204 |
| | Mopane | 0.78 | 8.27 ±1.32 | 0.198 ±0.071 | 0.976 ±0.009 | 2 | 0.80 ±0.03 | 3.96 ±1.70 | 0.629 ±1.043 | 0.841 ±0.272 | 3 | 0.050 | 0.550 | 0.481 |
| | Dung | 0.85 | 4.27 ±0.68 | 0.111 ±0.041 | 0.974 ±0.011 | 2 | 0.82 ±0.06 | 5.17 ±1.81 | 0.041 ±0.015 | 0.992 ±0.004 | 3 | 0.490 | 0.263 | 0.261 |
| | Savanna Grass | 0.75 | 6.34 ±1.00 | 0.068 ±0.026 | 0.989 ±0.004 | 3 | 0.71 | 5.48 | 0.019 | 0.997 | 1 | - | - | - |
| Photo-aged | Acacia | * | 11.65 ±1.87 | 0.349 ±0.119 | 0.970 ±0.011 | 2 | 0.81 ±0.08 | 4.73 ±0.94 | 0.284 ±0.323 | 0.940 ±0.069 | 2 | 0.134 | 0.835 | 0.653 |
| | Mopane | 0.78 | 12.32 ±1.96 | 0.295 ±0.102 | 0.976 ±0.009 | 2 | 0.76 | 4.02 | 0.110 | 0.973 | 1 | - | - | - |
| | Dung | 0.85 | 9.04 ±1.44 | 0.108 ±0.040 | 0.988 ±0.005 | 2 | 0.83 | 3.63 | 0.063 | 0.983 | 1 | - | - | - |



In cases where a direct comparison could be made (Wanza, mopane, dung, and savanna grass), MCE values were commensurate, so the combustion state was smoldering-dominated for both. A statistical analysis of MEC, MAC, and SSA values were performed using a two-tailed t-test with unequal variance; the results of which are in Table 5. Nearly all values could not be distinguished at a 95 % confidence interval – only the MEC of mopane had a P-value just under 0.05. The largest

difference between MAC values was for Wanza, which was larger in NCAT measurements by a factor of 9.3. Aside from this, MAC values were similar and agreed within a factor of 3.6. MAC values were generally larger for all NCAT experiments by a factor of 2.7 on average, with fresh measurements being larger by a factor of 3.3 and aged by a factor of 1.9. While these differences are not statistically significant, this could be due to a relatively small number of experiments. MEC values exhibited a better agreement, being within a factor of 3.1 at most and higher for NCAT measurements by a factor of 1.8 on average.

Unlike MAC, MEC values of aged aerosols showed greater differences than fresh ones. Upon aging, MEC values doubled, while MAC values increased by a factor of 1.3. While cross sections were generally higher for NCAT measurements, the resulting SSA values were nearly identical, for both fresh and aged aerosol. A potential explanation of this could lie, not in the optical measurement, but the mass measurement. The SMPS used at NCAT was limited to a maximum particle size of 710 nm, while LANL measurements had a wider range of instruments that could count particles between 10 nm and 10 µm. While

there may only be a small number of particles not included above the upper limit, they can contribute significantly to the total aerosol mass. This would produce an undercounting of particle mass, which would inflate the cross sections when they are calculated using Eq. 3.

    To corroborate this explanation, we examined the normalized size distributions of wanza and mopane, which are shown in Fig. S3. These were selected because the largest differences in optical properties were observed between laboratories for these

fuels. For both fuels, mass distrubtions in NCAT experiments appear to be truncated at 710 nm. Particles larger than the 710 nm accounted for ~0.1% of particle mass for wanza in LANL experiments. For mopane, 19 % by mass for burn #14, 24 % for burn #28, and 2 % for burn #36P were attributed to particles larger than 710 nm. While differences in the measured mass could not account for cross section differences for wanza, it could be a contributing effect for mopane and others.

## 6 Conclusions

We have measured the wavelength-dependent optical properties of BB aerosol from North American and sub-Saharan African fuels. We used a PASS as a reference instrument to calculate the multiple-scattering correction factor $C_\lambda$ of AE33 absorption measurements at 370, 470, and 520 nm. We found that there was no real dependence of $C_\lambda$ on MCE. When $C_\lambda$ was plotted against SSA, values slowly rose at low SSA values, followed by a sharp rise around an SSA of ~0.9. The trend of our $C_\lambda$ observations closely follow those of Yus-Díez et al. (2021) for $C_{637}$. $C_\lambda$ at other wavelengths were generally higher at all SSA

values and did not fit the functional form of $C_\lambda = A\omega/(1-\omega)+B$ particularly well, despite its previous use (Schmid et al., 2006; Yus-Díez et al., 2021). This is likely due to, at least in part, the to the scatter in the data and number of observations. Yus-Díez



et al. (2021) had a large number of ambient observations that they binned and averaged as a function of SSA before fitting those averages, whereas we have 31 burns with a total of 282 data points. A number of functions operating on either SSA or $C_\lambda$ were explored and the best functions of each are found in Table 4, along with the fit parameters at 370, 470, and 520 nm.

In terms of both $R^2$ and $X^2$, $-C_\lambda/(1-C_\lambda) = A\omega+B$ showed the best performance. This will provide an important parametrization of $C_\lambda$ specifically geared towards BB aerosol from African fuels under different aging states, and is of particular importance for future field work on the continent. Weighted mean values of $C_\lambda$ in this work were $C_{370} = 3.69$, $C_{470} = 5.65$, and $C_{520} = 6.39$, which was somewhat higher than several mean values proposed in previous works, though the same trend is observed for works where a trend is evident. This indicates that the multiple-scattering correction factor $C_\lambda$ in aethalometer measurements depends strongly on SSA and wavelength-dependent corrections are essential, particularly for $C_{370}$ that was different than $C_{470}$ and $C_{520}$. Since this dependence varies with aerosol composition, a fixed correction value is inadequate, especially in biomass burning scenarios.

An Ångström matrix plot of our observations produced ASE values ranging from -0.78 to 2.88 (mean of 0.61), while AAE ranged from 1.50 to 10.18 (mean of 6.47). AAE values were not unrealistically low, but the largest value observed here, for the smoldering combustion of Eucalyptus, has exceeded previous observations (Liu et al., 2018; Kirchstetter et al., 2004; Lack and Langridge, 2013). The Ångström matrix shows that African BB aerosol can have optical Ångström values more akin to dust. This shows that these fuels are distinct in their wavelength dependence from more typical BB aerosol.

We have previously demonstrated the flexibility of the tube furnace for generating BB aerosol under a variety of combustion conditions (Pokhrel et al., 2021b). To assess reproducibility between laboratories, a comparison was made between the MEC, MAC, and SSA of the same fuels combusted under smoldering-dominated conditions. Nearly all values could not be distinguished at a 95 % confidence interval. NCAT experiments were higher by a factor of 2.7 for MAC, on average, and 1.8 for MEC. There is evidence that these differences can be partially due to the more limited range of particle sizes measured at NCAT. Future work in cross section measurements should include fitting size distribution data with one or more log-normal distributions to account for missing high mass particles.

## 7 Author contributions

Marc N. Fiddler was involved in conceptualization, formal analysis, funding acquisition, supervision, visualization, draft writing, and editing. Vaios Moschos was involved in conceptualization, formal analysis, investigation, methodology, supervision, draft writing, and editing. Megan M. McRee was involved in formal analysis, investigation, methodology, validation, and editing. Abu Sayeed Md Shawon was involved in data curation, investigation, methodology, supervision, and editing. Kyle Gorkowski was involved in data curation, investigation, methodology, and supervision. James E. Lee was involved in data curation, methodology, and editing. Nevil A. Franco was involved in supervision and investigation. Katherine B. Benedict was involved in supervision, investigation, draft writing, and editing. Samir Kattel was involved in draft writing, formal analysis, and editing. Chelia Thompson was involved in formal analysis. Manvendra K. Dubey and Solomon Bililign



were involved in conceptualization, funding acquisition, project administration, resources, supervision, draft writing, and
editing.

## 8 Competing interests

The authors declare that they have no conflict of interest.

## 9 Acknowledgements

Fuels from Botswana were provided in collaboration with Gizaw Mengistu Tsidu at Botswana International University of
Science and Technology (BIUST). The fuels from Ethiopia were provided by Prof. Bekele at Addis Ababa University. Dr. Kip
Carrico's laboratory at New Mexico Tech performed measurements of fuel moisture content.
This manuscript has been approved for unlimited release by Los Alamos National Laboratory (LA-UR-25-25496)

## 10 Funding

The project was supported by funds from the Department of Energy under grant DE-FOA-0002688: Research Development
and Partnership Pilot (RDPP) program. The collaborative at Los Alamos was supported by the U.S. Department of Energy's
Atmospheric System Research, an Office of Science Biological and Environmental Research program; Los Alamos National
Laboratory is operated for the DOE by Triad National Security, LLC under contract F265. VM acknowledges support by the
Swiss National Science Foundation (SNSF) under the Postdoc Mobility Fellowship grant P500PN_210745.

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
