# Peer review of "Findings of the African Combustion Aerosol Collaborative Intercomparison Analysis (ACACIA) Pilot Project to Understand the Optical Properties of Biomass Burning Smoke"

_EGUsphere, 2025_

## Author Comment (AC1)

The study aims to optically characterize biomass burning aerosols from sub-Saharan African fuels, focusing on accurately determining the multiple-scattering correction factor for AE33 aethalometers and its relationship with particle single scattering albedo (SSA). The research develops a parametrization of the correction factor specific to African BB aerosols under different aging conditions, highlights their distinct wavelength dependence. I have following major questions for authors:

Our responses to the author will be in italic, while changes to the text will be in blue.

1. If emission data from different types of fuel combustion are fitted separately using your fit function, is there a large difference in the fitting quality? Is it possible that the fit works better for one or a few fuel types even if those types are not well-suited? Has the author considered this?

Authors Response: The reviewer raises in interesting point, particularly since a variety of fuels are presented. While we regard this as a strength for the enclosed work, there is likely value in examining a subset of measurements. We have added fuel types to Table 1 to enhance understanding of each fuel. We have also examined the fit results, looking at only emissions from woods found in Africa – the first four fuels in Table 1. Part of the motivation for part of this work is to use the AE33 in future long-term absorption measurements in Africa, so an African BB  $C_{\lambda}$  would be of practical use.

Changes to the Text: Starting at line 302, the text now reads "Of those,  $-C_{\lambda}/(1-C_{\lambda}) = A\omega + B$  performed the best, where 49 % of the variability of the Y-term is dependent on SSA and this equation had had the second lowest  $X^2$ . Solving for  $C_{\lambda}$ , this would have the form  $C_{\lambda} = (A\omega + B)/(A\omega + B - 1)$ , which has a strong potential for future use in SSA-based correction schemes, particularly where filter loading effects are minimized and optical properties dominate the measurement bias. While  $\arctan(C_{\lambda})$  had slightly lower  $X^2$  values, the  $R^2$  was also lower, so it exhibited a weaker  $C_{\lambda}$  dependence. The better  $X^2$  for  $\arctan(C_{\lambda})$  vs. SSA is mainly an artifact of all  $\arctan(C_{\lambda})$  flattening out.

Given the figures of merit for the AE33, mentioned in the introduction, it is likely to see use in long-term monitoring in future field campaigns in Africa. Supporting this, we have examined the above fit functions for woody African fuels (i.e. the first four fuels listed in Table 1), which was a subset of 20-21 data points for each wavelength. The results of this examination are in Table S2, and show general improvement in fit characteristics, with all  $X^2$  decreasing and all  $R^2$  increasing except the lowest three. The best fit function is still  $-C_{\lambda}/(1-C_{\lambda}) = A\omega + B$ , for the same reasons stated for the full data set. A fit of this function was also done for fresh, woody African fuels. The fit parameters of the best-performing fit equations that manipulate SSA (the nested exponential) and  $C_{\lambda}$  ( $-C_{\lambda}/(1-C_{\lambda})$ ) are shown in Table 4 and are plotted in Fig. S2 for all fuels, along with fit parameters for African woody fuels.

**Table 4.** The resulting fit parameters of functions applied to  $C_{\lambda}$  and SSA for the best overall fits. Parameters A and B are in the function and values are given at each wavelength in this study. Fit parameters are also given for just African woody fuels (fresh and aged) and only fresh African woods.

|                      |                                              | C370    |        | C470    |        | C520    |        |
|----------------------|----------------------------------------------|---------|--------|---------|--------|---------|--------|
| Fuels                | Function                                     | A       | В      | A       | В      | A       | В      |
| All                  | $C_{\lambda} = Ae^{e^{\omega}} + B$          | 0.3502  | 1.3030 | 0.2180  | 3.5103 | 0.4229  | 2.6202 |
| All                  | $-C_{\lambda}/(1-C_{\lambda}) = A\omega + B$ | -0.6074 | 1.7855 | -0.2489 | 1.4279 | -0.2972 | 1.4296 |
| African woods, all   | $-C_{\lambda}/(1-C_{\lambda}) = A\omega + B$ | -0.6425 | 1.8109 | -0.2579 | 1.4363 | -0.3116 | 1.4398 |
| African woods, fresh | $-C_{\lambda}/(1-C_{\lambda}) = A\omega + B$ | -0.7502 | 1.867  | -0.2804 | 1.4459 | -0.3255 | 1.4402 |

The following Table has also been added to SI:

**Table S2.** Fit functions applied to the plot of  $C_{\lambda}$  against SSA for African wood fuels (see Table 1) and the resulting  $\mathbb{R}^2$  and the Chi squared ( $\mathbb{X}^2$ ) values at each wavelength.

|                 |                                                                               |       | $\mathbb{R}^2$ |       |      | $X^2$ |       |
|-----------------|-------------------------------------------------------------------------------|-------|----------------|-------|------|-------|-------|
| Function        | Form                                                                          | 370   | 470            | 520   | 370  | 470   | 520   |
| Linear          | $C_{\lambda} = A\omega + B$                                                   | 0.467 | 0.311          | 0.393 | 4.12 | 5.21  | 9.73  |
| Polynomial      | $C_{\lambda} = A\omega^2 + B\omega + D$                                       | 0.491 | 0.322          | 0.396 | 3.99 | 5.10  | 9.71  |
| Log             | $C_{\lambda} = A \cdot \ln(\omega) + B$                                       | 0.446 | 0.318          | 0.386 | 4.33 | 5.14  | 9.88  |
| Exponential     | $C_{\lambda} = Ae^{B\omega}$                                                  | 0.485 | 0.305          | 0.396 | 4.11 | 5.39  | 10.04 |
| Power Law       | $C_{\lambda} = A\omega^{B}$                                                   | 0.463 | 0.314          | 0.394 | 4.30 | 5.29  | 10.06 |
| Schmid/Yus-Díez | $C_{\lambda} = A\omega/(1-\omega)+B$                                          | 0.433 | 0.329          | 0.379 | 5.06 | 5.53  | 11.08 |
|                 | $C_{\lambda} = -A/\ln(\omega) + B$                                            | 0.434 | 0.330          | 0.379 | 5.06 | 5.53  | 11.07 |
| Arctangent      | $C_{\lambda} = A \cdot \arctan(\omega) + B$                                   | 0.453 | 0.317          | 0.389 | 4.26 | 5.15  | 9.81  |
|                 | $C_{\lambda} = \mathbf{A} \cdot e^{(\omega - 1)} / (1 - \omega) + \mathbf{B}$ | 0.430 | 0.329          | 0.379 | 5.09 | 5.54  | 11.08 |
| Nested Exp.     | $C_{\lambda} = \mathbf{A} \cdot e^{e^{\omega}} + \mathbf{B}$                  | 0.511 | 0.284          | 0.395 | 3.85 | 5.51  | 9.85  |
|                 | $-C_{\lambda}/(1-C_{\lambda}) = A\omega + B$                                  | 0.576 | 0.429          | 0.476 | 0.14 | 0.04  | 0.05  |
|                 | $1/\ln(C_{\lambda}) = A\omega + B$                                            | 0.574 | 0.417          | 0.470 | 0.37 | 0.16  | 0.20  |
|                 | $\arctan(C_{\lambda}) = A\omega + B$                                          | 0.562 | 0.408          | 0.469 | 0.03 | 0.02  | 0.02  |

2. The paper mentions that PAM was used to simulate aging experiments. Specifically, what degree of aging equivalent 3days? or 7days? did the authors simulate? During the aging simulation, did the degree of aging vary?

Authors Response: We thank the reviewer for pointing out this omission. With the exception of part of one experiment, there was only a little variation.

Changes to the Text: The following was added to line 124: Several experiments used a potential aerosol mass (PAM) flow reactor, which was operated with two lamps. BB aerosol experienced 2.2 equivalent days of OH oxidation in burn 35O, 1.9 days in 36O and 42O, and most of burn 39O had the equivalent of 2.2 days except for the first 9 minutes where there were 5.0 equivalent days of oxidation. Except burn 39O, the equivalent OH oxidation time had a standard deviation of less than 40 min.

Moreover, the authors combined fresh and aged data in the linear relationship shown in Fig. 3, which makes it difficult to see the differences in SSA correlation between fresh and aged emissions. Therefore, it is unclear whether the authors' statement that the results also apply to aged aerosols is justified. It is recommended that the authors present separate linear fit plots of the fit function for fresh and aged data.

Authors Response: The authors admit that the aged measurements should be differentiated from fresh ones. We have done this in Figure 3. We have found that the best fit function for fresh measurements is still  $-C_{\lambda}/(1-C_{\lambda}) = A\omega + B$ , as we discuss in response to the previous question. There are only four data points at each wavelength for aged fuels. Since these are all at high SSA values, they are clustered to the right-hand side of the plot and there are so few data points, a linear fit is quite different and different fit functions cannot be distinguished. For these reasons, the authors feel that fitting only aged samples is not useful, though the values for these points are given in SI for the reader to use.

Changes to the Text: Figure 3 is now:

Figure 3. The aethalometer correction factor  $C_{\lambda}$  plotted against SSA at three wavelengths. PAM oxidation experiments were included and are marked with black dots. Results of a linear fit are shown.

3. First, Fig. 4 is difficult to interpret because the dashed lines are too cluttered. Second, what do the shaded areas represent? Does the pink shading indicate aged aerosols and the grey shading indicate fresh aerosols? From my understanding, there is still considerable discrepancy between the experimental data and the reference data. Since the comparison is made for similar sources, why do the authors' experimental results differ so much from previous studies in Africa?

Authors Response: The dashed lines have been replaced by solid lines. Unfortunately, we cannot change the crowding of these lines because they've been established by previous observations in other works. While we stand by our observations, it is clear that differences between these sets of data require more explanation. We would not call these previous measurements "reference data" since they were simply previous measurements of similar African fuels in our lab. Since differences have been observed, we will attempt to explain it in the text.

Changes to the Text: Figure 4 has been changed:

Figure 4. The Ångström matrix plot (ASE vs AAE) for fresh (black dot) and PAM-aged (tan dot) observations in this work. Previous measurements on similar fuels are shown in shaded areas for fresh and photo-aged BB aerosol (McRee et al. 2024).

The following has been added to the end of section 4:

This clearly shows that BB aerosol from African fuel sources are distinct in their optical properties. Differences between observations in this work and studies McRee et al. (2024) are likely due to a number of factors, including differences in wavelength range and instrumentation; 405 and 532 nm scattering and absorption measurements using a PASS in this work vs. previous absorption measurements at 520 and 590 nm using an AE33 and scattering measurements at 453 and 554 nm with a nephelometer. The correction for the AE33 used a different correction method (Moschos et al., 2024) and a single power-law relationship may not hold so close to the UV. It is

also very likely that there are differences in photo-aging between the PAM and smog chamber. The largest difference between studies is that McRee et al. (2024) focused only on smoldering-dominated combustion, which would have a relatively high BrC content, while a variety of combustion states were explored in this work. Regardless, in both studies, the range of values for both AAE and ASE decreased upon photoaging, as well as with dark aging and dark aging with additional nitrate radical (McRee et al., 2024). This demonstrated that both processes reduced the wavelength dependence of scattering and absorption.

If AAE and ASE differ substantially, could this affect the general applicability of the fit function to African fuel data?

Authors Response: Data in Figure 4 was derived purely from the PASS, so the fit functions used for the AE33 do not factor into this plot.

---

## Author Comment (AC2)

We thank the reviewer for their suggestions and efforts to strengthen the paper. The author's responses are in *italics*, while changes to the text are in blue text.

Reviewer #2 comments

This manuscript by Fiddler et al. presents the findings of the ACACIA pilot project, which aimed to characterize the optical properties of BB aerosols from African fuels and to intercompare measurement techniques (AE33 aethalometer vs. photoacoustic reference). The study is significant and timely: Africa is a major source of BB aerosol with growing impact on climate and air quality, yet its aerosol optical properties are less studied. The work fits well within the scope of Atmospheric Measurement Techniques, as it deals with instrument calibration, intercomparison, and methodological advances in aerosol optical measurements. Most of my criticisms are relatively major and should be straightforward to address. I believe the paper will be a valuable contribution to AMT after the authors revise the manuscript in response to the specific comments below.

1. The use of a three-wavelength photoacoustic spectrometer (PASS-3) as the reference for AE33 absorption measurements is appropriate. However, since the PASS-3 provided data at 405 nm and 532 nm (with the 781 nm channel not used) while the AE33 has channels at 370, 470, 520 nm, the authors extrapolate or interpolate the absorption to those wavelengths (likely assuming a power-law wavelength dependence via an Ångström exponent). Please clarify in the methods how this extrapolation was done and discuss the associated uncertainty. Figure S1 (burn 17 example) is mentioned for extrapolating using AAE – the manuscript should ensure the procedure is described in Section 2.4. How sensitive are the derived $C\lambda$ values to the assumed power-law? For example, if the aerosol absorption does not follow a strict power-law between 405 and 532 nm, could that bias the extrapolated $\alpha\_abs$ at 370 or 470 nm? Since the PASS measurement uncertainty is noted as ~40%, it would also be useful to comment on how this uncertainty propagates to the reported $C\lambda$. Right now the paper reports variability (standard deviations) of $C\lambda$ across runs, but a statement on the absolute accuracy or uncertainty of the $C\lambda$ values (accounting for instrument calibration uncertainties) would strengthen confidence in the results.

*Author Response: We have included text pertaining to the interpolation/extrapolation was done. Given that there are reference measurements at only two wavelengths, the resulting values of $C_\lambda$ are expected to be highly sensitive, which is mainly mathematical statement, not limitation of this technique or these measurements. We believe a power-law assumption is best suited for this work, based on previous observations, with the understanding that there are limitations to this model. However, we would be unable to evaluate the quality of the power-law relationship vs. some other relationship with the data at hand.*

The following was added to line 182:

Since the PASS operates at other wavelengths (405 and 532 nm), $\alpha abs,\lambda$ at AE33 wavelengths of 370, 470, and 520 nm were extrapolated by assuming a power-law relationship (i.e. deriving an Ångström absorption exponent (AAE)). This power-law relationship has been used in absorption

measurements to account for spectral differences between different instruments (Arnott et al., 2005; Collaud Coen et al., 2010). This extrapolation/interpolation is shown in Fig. S1 for burn 17. Extrapolated values of $\alpha_{abs,\lambda}$ were derived from a $\log_{10}$-$\log_{10}$ graph of measured $\alpha_{abs,\lambda}$ values. Using the power-law equation $\alpha_{abs,\lambda} = m\lambda^b$, $\alpha_{abs,\lambda}$ was calculated, where $m$ is $10^{intercept}$ and $b$ is the slope of the $\log_{10}$-$\log_{10}$ graph.

*Author Response: Regarding the propagation of uncertainty in calculating $C_\lambda$, we have addressed that on lines 212.*

Changes to the text: The remaining 31 burns had an average RSD of 6.1 % for $C_\lambda$ and 0.4 % for SSA over all wavelengths. This is smaller than the 25 % RSD found previously (Moschos et al., 2024). Previous $\alpha_{atn,\lambda}$ measurements with the AE33 showed a repeatability uncertainty of 15-30 %, while the resulting AAE uncertainty was <5 % (Moschos et al., 2024). That is, the ratio of $\alpha_{atn,\lambda}$ between different wavelengths is very consistent between burns. When propagating the AE33 (15-30 %) and PASS uncertainty (40 % and assuming the RSD of extrapolated/interpolated points are the same) in quadrature, a calculated RSD of 43-50 % is expected for $C_\lambda$.

2. Tables 3 and 4 contain many functional forms, which makes the findings hard to follow. The manuscript should down-select and focus on the best-performing function ($-C\lambda/(1-C\lambda) = A\cdot\omega + B$), highlights its advantages, and discuss its potential physical meaning. Otherwise, it may appear overly data-driven without mechanistic insight.

*Author Response: The authors explored a number of equations that produce relatively sharp transitions. This included squareplus, mish, softplus, generalized logistic, and the error functional. These performed poorly and were not included in the presented work. While there may be several equations in Tables 3 and 4, the authors have already kept these to a minimum. Table 4 has been revised, according to another reviewer's request:*

**Table 4.** The resulting fit parameters of functions applied to $C_\lambda$ and SSA for the best overall fits. Parameters A and B are in the function and values are given at each wavelength in this study. Fit parameters are also given for just African woody fuels (fresh and aged) and only fresh African woods.

| Fuels | Function | $C_{370}$ A | $C_{370}$ B | $C_{470}$ A | $C_{470}$ B | $C_{520}$ A | $C_{520}$ B |
|---|---|---|---|---|---|---|---|
| All | $C_\lambda = Ae^{e^\omega} + B$ | 0.3502 | 1.3030 | 0.2180 | 3.5103 | 0.4229 | 2.6202 |
| All | $-C_\lambda/(1-C_\lambda) = A\omega + B$ | -0.6074 | 1.7855 | -0.2489 | 1.4279 | -0.2972 | 1.4296 |
| African woods, all | $-C_\lambda/(1-C_\lambda) = A\omega + B$ | -0.6425 | 1.8109 | -0.2579 | 1.4363 | -0.3116 | 1.4398 |
| African woods, fresh | $-C_\lambda/(1-C_\lambda) = A\omega + B$ | -0.7502 | 1.867 | -0.2804 | 1.4459 | -0.3255 | 1.4402 |

*Regarding the physical meaning of the $-C_\lambda/(1-C_\lambda) = A\omega + B$ function, this is difficult to address. While Schmid and Yus-Díez based their function on underlying scattering theory, the form of their function also doesn't behave as well as the one found in this work.*

3. The reported AAE values above 10 are unusual and exceed most prior studies. The manuscript should provide a more careful discussion of the possible causes. Without this,

readers may question the reliability of these extreme values. On the other hand, negative or very low ASE might indicate presence of super-micron particles or ash that scatter more efficiently at longer wavelengths – perhaps related to soil/dust or inorganic residue in the fuel (especially dung might contain soil, and savannah grass fires could entrain dust). A short discussion linking these observations to possible physical causes would strengthen the impact. Currently the manuscript states the observation (dust-like optical values) but does not delve into why. Even if a detailed chemical analysis is outside the scope, a sentence like "These extreme Ångström exponent values may result from the high fraction of smoldering organic carbon (leading to very strong wavelength-dependent absorption) and/or the presence of coarse ash particles (leading to anomalous scattering spectra), distinguishing African BB aerosol from typical forest-fire smoke" would be helpful. Additionally, please clarify in the figure/caption how the reader should interpret negative ASE values – some readers might be unfamiliar with the idea that a negative exponent is possible. It might be worth noting that this occurs when larger particle modes cause scattering to increase with wavelength in the measured range (or note if it is within experimental uncertainty). This clarification can prevent confusion.

*Author Response: The authors agree that further explanation is warranted and thank them for their suggestions and insight. The cyclone mentioned below has been added to Fig. 1, shown here:*

[Figure]

**Figure 1. A diagram of the system at LANL used for generating and characterizing BB aerosol.**

To line 342, we have revised the text:

Values observed in this work were wide ranging. ASE ranged from -0.78 to 2.88 with an average of 0.61, where negative values result in an MSC that is greater at longer wavelengths. These

negative ASE values may result from the presence of super-micron particles that have higher scattering efficiencies at longer wavelengths. This can be seen in the oscillating portion of Fig. 11.5b in Bohren and Huffman (1998). Such large particles are present in some burns, as shown in Fig. S3.

To line 353, we have added:

… While these literature values were close to the range observed for similar African fuels and previous studies (McRee et al., 2024), as indicated in Fig. 4, the range of values in this work is clearly much greater and even slightly exceeding previously observed values of AAE, being more akin to dust observations. This clearly shows that BB aerosol from African fuel sources are distinct in their optical properties. Such large Ångström exponent values may result from the high fraction of smoldering organic carbon (leading to very strong wavelength-dependent absorption), which distinguishes African BB aerosol from typical forest-fire smoke. Clearly, characterizing the chemical composition of these BB aerosol is also important for explaining these observations. The presence of coarse ash particles (leading to anomalous scattering spectra) could also produce these extreme AAE, though this is unlikely in this system, since such ash would be removed by a cyclone with a cut point of 2.5 µm placed after the mixing tank (see Fig. 1).

Differences between observations in this work and studies McRee et al. (2024) are likely due to a number of factors, including differences in wavelength range and instrumentation; 405 and 532 nm scattering and absorption measurements using a PASS in this work vs. previous absorption measurements at 520 and 590 nm using an AE33 and scattering measurements at 453 and 554 nm with a nephelometer. The correction for the AE33 used a different correction method (Moschos et al., 2024) and a single power-law relationship may not hold so close to the UV. It is also very likely that there are differences in photo-aging between the PAM and smog chamber. The largest difference between studies is that McRee et al. (2024) focused only on smoldering-dominated combustion, which would have a relatively high BrC content, while a variety of combustion states were explored in this work. Regardless, in both studies, the range of values for both AAE and ASE decreased upon photoaging, as well as with dark aging and dark aging with additional nitrate radical (McRee et al., 2024). This demonstrated that both processes reduced the wavelength dependence of scattering and absorption.

4. Table 5 shows MAC discrepancies as large as a factor of 9, which cannot be fully explained by the SMPS upper size cut alone. A dedicated subsection on measurement uncertainties and potential systematic bias would improve transparency and address concerns regarding reproducibility.

*Author Response: As stated, it was only Wanza that had such a large difference. All other MAC values were within a factor of 3.6. That being said, further explanation is warranted.*

To line 398, we have added: For mopane, 19 % by mass for burn #14, 24 % for burn #28, and 2 % for burn #36P were attributed to particles larger than 710 nm. In addition, two lognormal distributions were fit to NCAT experiments, where the total mass agreed within <1.1 %. From this analysis, 4.9 % of particle mass exceeded 710 nm for wanza, and 17.3 % for mopane. While differences in the measured mass could not account for cross section differences for wanza, it

could be a contributing effect for mopane and others. Additionally, the AE33-based MAC has a propagated uncertainty of 18-32 % based on a particle counting uncertainty of 10 %, a CRDS-based MEC has an 11 % uncertainty (Singh et al., 2014), while the PASS-based MAC and MSC would be 41%. As such, random errors could contribute to these observed differences. Differences could also be the result of mixing state of the aerosol, as there are differences in dilution, cooling, preferential wall losses, and residence time between the 9 m³ chamber and the 34 L mixing tank. Differences in the size distribution are apparent (Fig. S3), which is likely a result of these factors. Given that the SSA results for LANL and NCAT are nearly the same, the balance between scattering and absorption is very robust, despite differences in observed cross sections.

Minor comments:

1. In figures 2 and 3, pls add 95% confidence bands around fit lines.

*Author Response: For Figure 2, we believe the included $R^2$ values are sufficient to demonstrate that $C_\lambda$ correlates poorly with MCE. For Figure 3, this would be very busy and, regardless, a linear fit is not the best function to fit. We have presented the best performing functions, which are now in Figure S2 and is included here:*

[Figure]

**Figure S2.** Results of the best fit functions of $C_\lambda$ to SSA for all fuels in this study: (a) the nested exponential of SSA vs. $C_\lambda$ and (b) SSA vs. $- C_\lambda(1-C_\lambda)$.

2. Use color coding to distinguish fuels, which aid readability.

*Author Response: Points in the forementioned figures are already color coded by wavelength. Since we have differentiated woody African fuels from others elsewhere in this work, we have revised Figure 3 to include symbols for African woody fuels, other fuels, and aged experiments. Should the reader wish to further parse the data, it will all be available for each figure in an SI table. It is as follows:*

[Figure]

**Figure 3. The aethalometer correction factor $C_\lambda$ plotted against SSA at three wavelengths. PAM oxidation experiments were included and are marked with black dots. Results of a linear fit are shown.**

3. The NCAT combustion chamber description is too brief and only references McRee et al. (2024). Pls summarize key features, such as chamber volume, RH control, oxidant usage.

*Author Response: We have included the characterization paper for the chamber (Smith et al. 2019) and the most recent and relevant paper for the current chamber configuration (McRee et al. 2024). While the addition of oxidants and water was performed in McRee et al., that data was not used in this work, so detailed information on RH control and introduction of oxidants is not germane to this paper.*

We have modified the text to read "While oxidants and water can be added to the chamber (McRee et al., 2024; Mouton et al., 2023), only fresh emissions without the addition of more

oxidant were studied in this work. The chamber was kept dry for these experiments, where the RH was 0–10 %." *on line 152. The chamber volume was stated earlier.*

4.  When referring to figures in the text, use consistent style (e.g., "Fig. 4" vs "Figure 4"). It looks like the manuscript mostly uses "Figure" spelled out, which is fine. Just ensure each figure is called out in order. I noticed Figure 1 is not explicitly referenced in the portions I read (it might be referenced in Section 2.2 or 2.3 when describing the setups – if not, please include a reference to Fig. 1 in the text so readers know to look at the schematic/configuration).

*Author Response: This has been corrected throughout the document. Fig. 1 is referenced on line 117 in the revised manuscript.*

5.  In Equation formatting, make sure all variables are defined. Equation (3) defines MAC_λ = α_abs,λ / M – earlier in the text, define "M" as the particulate mass concentration (µg m$^{-3}$) if not already done. Likewise, if "MEC" (mass extinction cross-section) is used as a term, define it clearly on first use (I believe it is defined around Eq. 2, but just to be certain).

*Author Response: These have been defined before Eq. 3. On line 189 we state that* "Cross sections were calculated using the particle mass loading $M$ (µg m$^{-3}$)… The mass absorption cross-section (MAC, m$^2$ g$^{-1}$) is calculated using Eq. 3, with mass scattering and extinction cross-sections (MSC and MEC, respectively) being calculated similarly." *We have made sure that all variables have been described.*